# Sensing and Signaling of Methionine Metabolism

**DOI:** 10.3390/metabo11020083

**Published:** 2021-01-31

**Authors:** Linda Lauinger, Peter Kaiser

**Affiliations:** Department of Biological Chemistry, School of Medicine, University of California, Irvine, CA 92697, USA; llauinge@uci.edu

**Keywords:** methionine, S-adenosylmethionine (SAM), methionine/SAM sensing, cancer, aging, cell cycle

## Abstract

Availability of the amino acid methionine shows remarkable effects on the physiology of individual cells and whole organisms. For example, most cancer cells, but not normal cells, are hyper dependent on high flux through metabolic pathways connected to methionine, and diets restricted for methionine increase healthy lifespan in model organisms. Methionine’s impact on physiology goes beyond its role in initiation of translation and incorporation in proteins. Many of its metabolites have a major influence on cellular functions including epigenetic regulation, maintenance of redox balance, polyamine synthesis, and phospholipid homeostasis. As a central component of such essential pathways, cells require mechanisms to sense methionine availability. When methionine levels are low, cellular response programs induce transcriptional and signaling states to remodel metabolic programs and maintain methionine metabolism. In addition, an evolutionary conserved cell cycle arrest is induced to ensure cellular and genomic integrity during methionine starvation conditions. Methionine and its metabolites are critical for cell growth, proliferation, and development in all organisms. However, mechanisms of methionine perception are diverse. Here we review current knowledge about mechanisms of methionine sensing in yeast and mammalian cells, and will discuss the impact of methionine imbalance on cancer and aging.

## 1. Introduction

Cells must be able to properly assess their environment for the availability of resources required for growth. Hence, systems that sense the abundance of essential nutrients are needed to signal a potential supply deficit to coordinate metabolism and growth. The sulfur containing amino acid methionine presents a key metabolite with major influence on translation, epigenetics, cell proliferation, and various signaling cascades. Sensing the status of methionine metabolism and some of its metabolites is thus critical for cellular, epigenetic, and genomic integrity. In most known pathways S-adenosylmethionine (SAM) levels, rather than methionine, seem to be sensed to monitor the status of methionine metabolism. Epigenetic regulation, nucleotide biosynthesis, and membrane lipid homeostasis depend on the abundance of SAM, a key product of methionine metabolism and the major methyl-group donor in cells. When levels of methionine or SAM are low, cellular response programs are activated to combat these conditions. Induction of methionine transporters and upregulation of de novo methionine biosynthesis in yeast, as well as autophagy are ways to restore levels of methionine and its metabolites. In addition, a cell cycle arrest is observed and if cells are unable to recover from these nutrient stress conditions, apoptosis is induced to protect the organism from cells that could escape their epigenetic identity due to reduced ability to maintain chromatin methylation marks. The general principle of methionine as a growth signal seems to be conserved throughout different organisms, however the mechanisms for methionine perception are different. Here we summarize our current understanding of how methionine or SAM is perceived in yeast and mammalian cells. We will examine the mechanisms cells use to sense and respond to methionine starvation conditions, how cells try to overcome them to ensure organismal health, and the significance of methionine and SAM in the context of aging and cancer.

## 2. Methionine Metabolism in Yeast and Mammalian Cells

Beyond initiating and sustaining translation, methionine plays key roles in epigenetic regulations (chromatin methylation), nuclear functions (polyamines), and in maintaining the redox status in cells (glutathione). Methionine is rapidly converted into the central one-carbon cycle metabolite S-adenosylmethionine (SAM, sometimes referred to as AdoMet). Adenosine from ATP is transferred to the sulfur in methionine by methionine adenosyl transferase (MAT2A) in mammals or its homologs in yeast, SAM synthetase (SAM1/SAM2). SAM is the primary methyl group donor for methylation events of DNA, proteins, and lipids. After the transfer of the methyl group during methylation reactions, S-adenosylhomocysteine (SAH) remains, which is then hydrolyzed into adenosine and homocysteine by SAH hydrolase. Homocysteine methyltransferase (HMT) remethylates homocysteine to complete the methionine cycle. Homocysteine is remethylated using 5-methyltetrahydrofolate (5-MTHF) as the methyl donor, connecting the folate cycle to generation of methionine from homocysteine. In addition to 5-MTHF, Vitamin B_12_ (cobalamin) is required for the re-methylation step in mammals. Yeast HMTs are cobalamin-independent enzymes [1,2] (Figure 1; in green).

Next to generating methylation potential, methionine is also indirectly involved in the synthesis of polyamines. Polyamines are essential for cell growth and require decarboxylated SAM (dcSAM) as the donor for aminopropyl groups for their synthesis [3]. After aminopropyl group transfer, dcSAM is converted into 5′-deoxy-5′methylthioadenosine (MTA), which is subsequently recycled through multiple steps in the methionine salvage pathway to regenerate adenine and methionine (Figure 1; in red).

Methionine also contributes to maintain the redox status in cells by supplying the junction metabolite homocysteine as a substrate to the transsulfuration pathway [4,5]. Through cystathionine, homocysteine can be converted to cysteine [6], which is incorporated into the antioxidants taurine and glutathione (GSH). When GSH is scavenging reactive oxygen species (ROS) it is reversibly oxidized to GSSG. Notably, conversion of homocysteine to cysteine is unidirectional in mammals [7], yeast however is able to convert cysteine into homocysteine in a reversed reaction [8]. The importance of methionine metabolism for glutathione synthesis is evident in yeast where transcription programs responsive to methionine availability include genes encoding enzymes for glutathione synthesis [8], whereby oxidative stress and methionine starvation induce very similar gene expression programs (Figure 1; in blue).

In mammals, methionine is an essential amino acid, hence it needs to be obtained through the diet or supplemented to cultured cells via the growth media. In most fungi however, methionine can be generated through the sulfate assimilation pathway. In an energy-demanding (2 ATP and 4 NADPH molecules) enzyme cascade, sulfate is reduced to sulfite and eventually to sulfide [8]. Homocysteine, the precursor for methionine, is then generated by sulfide incorporation into a carbon backbone derived from homoserine [8,9]. As mentioned above, homocysteine can then be methylated with components of the folate cycle to generate methionine (Figure 1; in yellow).

## 3. Methionine Perception in Yeast

Organisms can adapt to a broad spectrum of environmental conditions, in which sudden and dramatic changes are perceived as stress. These environmental cues trigger specific response programs that facilitate survival and eventually adaptation to the new conditions. Methionine limitation in yeast invokes a complex transcriptional response. Most of these induced genes (MET genes) depend on the basic leucine zipper (bZIP) type transcriptional activator Met4 [10]. Met4 induces MET gene expression when sulfur containing amino acids are limiting, while under normal growth conditions Met4 is kept in a transcriptionally inactive state by the ubiquitin ligase SCF^Met30^ (Skp1, Cullins, F-box proteins) [11]. In yeast, SCF^Met30^ coordinates one major methionine sensing pathway that orchestrates cellular response programs during nutritional and heavy metal stress. As mentioned above, anti-oxidant and methionine response pathways have similar transcriptional outputs and overlapping metabolic pathways that are coordinated by the transcription factor Met4. SCF^Met30^ senses both methionine levels and heavy metals, but mechanisms of stress perception are profoundly different [12,13,14,15] (Figure 2A; blue box).

The two most critical substrates of the SCF^Met30^ ubiquitin ligase are the aforementioned transcriptional activator Met4 and the cell cycle inhibitor Met32. When intracellular methionine is plentiful, SCF^Met30^ catalyzes the ubiquitylation of Met4. Methionine or a methionine metabolite are required for the interaction between SCF^Met30^ and its substrates Met4 and Met32 [12] (Figure 2C; yellow box). The exact mechanism how methionine mediates the Met30/Met4 interaction has not been elucidated, but this ubiquitin ligase/substrate binding is clearly sensitive to intracellular methionine levels and responds rapidly to metabolite changes. Intriguingly, Met4 is kept in a transcriptionally inactive state by the post-translational poly-ubiquitin modification at lysine residue 163 [16]. Even though the attached K48 ubiquitin chain is a canonical degradation signal, Met4 is not recruited to the 26S proteasome for degradation (Figure 2B; green box). Met4 contains two internal ubiquitin binding domains (UBD) that bind the K48 chain with high affinity and shield it from 26S proteasomal recognition [17,18,19]. The tandem UBD also serves as a transactivation domain and was shown to facilitate interaction with the mediator, a transcriptional coactivator complex [19]. It was demonstrated that binding of the ubiquitin chain prevents interaction with mediator complex and thus keeps Met4 transcriptionally inactive. During methionine starvation conditions, when SAM levels are low, the ubiquitin chain topology on Met4 switches from K48 to K11 enriched chain types. K11 linkages do not compete with mediator binding, thus allowing the initiation of Met4 activation. Initiation is followed by mediator binding, deubiquitylation of Met4, and the start of the transcriptional response program to restore sulfur containing metabolites (Figure 2D; red box). Once methionine and SAM levels are restored, SCF^Met30^ binding to Met4 is promoted resulting in its ubiquitylation and return to a stable, but transcriptionally inactive state. The mechanism of non-proteolytic ubiquitylation provides a pool of readily available Met4, which can be quickly activated to respond to nutritional stress.

Met4 also functions as a substrate adaptor within the SCF^Met30/Met4^ ubiquitin ligase for the cell cycle inhibitor Met32. Thereby, methionine levels indirectly control Met32 ubiquitylation through modulating binding of the Met4/Met32 complex to Met30 and allow integration of cell cycle control with methionine metabolism [12]. Under normal growth conditions, the transcriptional activator recruits Met32 to SCF^Met30^ to enable its ubiquitylation and subsequent degradation to ensure cell cycle progression. Accordingly, under low methionine conditions, Met4 loses its interaction to the ubiquitin ligase and Met32 is no longer recruited for ubiquitylation. This results in the stabilization and accumulation of Met32, which in turn triggers a cell cycle arrest. When methionine levels are restored through the Met4-mediated transcriptional response, the arrest is released and proliferation can continue [16,18,20]. The SCF^Met30^ controlled methionine sensing pathway is the major pathway that coordinates cell proliferation with methionine metabolism in yeast. It thereby ensures epigenetic stability by preventing cells to enter S-phase during conditions when methionine-derived metabolites, such as SAM, are too low to ensure efficient duplication of epigenetic methylation marks. The mammalian homolog of SCF^Met30^ is SCF^βTRCP^. To the best of our knowledge, a possible involvement of SCF^βTRCP^ in coordinating cell proliferation with methionine metabolism has so far not been investigated.

A second yeast signaling pathway that responds to changes in methionine and SAM levels was described by Tu and colleagues in which they demonstrate a link between methylation potential and the regulation of autophagy [21]. Next to regulating systematic recycling and degradation of cellular components through the lysosome, autophagy also enables cells to adapt to changes in their metabolic state [22,23]. A key regulatory element that controls this process is the nutrient-sensitive Target of Rapamycin Complex 1 (TORC1) [23]. The catalytic center of the TORC1 complex is the serine/threonine protein kinase target of rapamycin (TOR). This highly conserved enzyme is a member of the phosphoinositide 3-kinase (PI3K)-related kinase family [24]. During sufficient nutrient availability, TORC1 is active and catalyzes downstream phosphorylation events that inhibit autophagy and promote growth [25]. TORC1 activity can be regulated on multiple levels by different factors depending on environmental conditions [26]. A decrease in availability of certain amino acids results in TORC1 inactivation, hence triggering autophagy [27]. Both mammals and yeast engage TORC1 in sensing of methionine metabolism, albeit by very different mechanisms (see below for the mammalian system). In yeast, protein phosphatase 2A (PP2A) plays a key role. The PP2A holoenzyme consists of the PP2Aa scaffold subunit, which forms the assembly platform for the catalytic PP2Ac and one of several substrate-selective PP2Ab subunits. Interestingly, the C-terminus of the catalytic PP2Ac subunit is carboxy-methylated at the carboxy-terminus [28]. Under normal growth conditions the two yeast PP2Ac subunits, Pph21 and Pph22, are methylated by Ppm1p and promote dephosphorylation of Nrp2p, a component of an autophagy repressing protein complex SEACIT [21]. Dephosphorylated Nrp2p is active and effectively prevents autophagy (Figure 3). Low methionine levels result in decreased SAM/SAH ratio, a measure of the cellular methylation potential [21]. These conditions result in demethylated PP2Ac and therewith inactivated phosphatase activity. Consequently, Nrp2p accumulates in the inactive phosphorylated form and no longer suppresses autophagy. These results describe an elegant mechanism by which yeast cells sense methionine metabolism through monitoring the intracellular methylation potential, and induce autophagy to re-feed metabolism [21,29,30].

In a more recent study by Tu and colleagues, they report a second key metabolic adaptation downstream of demethylated PP2Ac [31]. Decreased PP2A activity during methionine starvation results in hyperphosphorylation of histone demethylases, which increases chromatin binding and activity, thereby enhancing demethylation and preventing remethylation. This process also increases SAH, which is a potent inhibitor of methyltransferases [32]. Overall, these mechanisms result in SAM preservation through limiting its consumption by methyltransferases [31].

Methionine availability also influences the thiolation of certain tRNAs to regulate cellular translation capacity and metabolic homeostasis [30,33,34]. Interestingly, the aforementioned autophagy repressing complex SEACIT containing Npr2p also negatively regulates tRNA thiolation [30], Hence interlinking cell growth control and translational capacity to spare sulfur containing amino acids during methionine starvation. 

A remarkable mechanism for sulfur sparing has also been observed in yeast during cadmium exposure, further emphasizing the intimate link between methionine metabolism and antioxidant homeostasis. Yeast cells exposed to cadmium shift expression of several abundant metabolic enzymes to isozymes that are lacking methionine or cysteine residues, thereby reducing methionine and cysteine consumption to maintain glutathione production and reduce oxidative vulnerabilities of key enzymes [35].

Methionine perception in yeast occurs at several levels, highlighting the central importance and intersection of this metabolic pathway with a plethora of cell functions. So far, no elements have been identified that actively bind methionine, SAM, or other related metabolites to directly measure their levels. The SCF^Met30^ system is a potential candidate for such metabolite binding elements, but other systems appear to monitor methionine metabolism indirectly by measuring the cellular methylation potential, which is determined by the SAM/SAH ratio. As outlined in the next chapter, mammals follow a more direct sensing mechanism, which involves the SAM binding effector SAMTOR to integrate methionine metabolism with TORC1 activity [36].

## 4. Methionine Perception in Mammals

In mammalian cells, the serine/threonine kinase mTOR was shown to be a central factor for the sensing of methionine and its derivates. Similar to yeast, mTOR is also the catalytic center of the two multi-protein complexes, mTORC1 and mTORC2. Even though both mTORC complexes share their catalytic subunit, they regulate distinct functions in the cell. Cell growth is mainly controlled by mTORC1 activity, whereas cell survival and proliferation are coordinated via mTORC2 [37]. To control cell growth, cells need to associate the abundance of growth factors and availability of nutrients in their environment with the production of biomass [38]. mTORC1 is a crucial component in the nutrient sensing pathway and its activity is regulated through the nucleotide state of lysosomal Rag GTPases. Not only lysosomal but also cytosolic amino acid availability is sensed via distinct mechanisms with mTORC1-associated multi-component complexes being the central elements that regulate Rag GTPase activity [24]. A negative regulator in this pathway is the GTPase activation protein complex GATOR1 (SEACIT in yeast) [39]. To interact with the Rag GTPases, GATOR1 requires lysosomal localization, which is facilitated by the scaffold protein complex KICSTOR [40,41]. GATOR2, a positive regulator of mTORC1 is an additional lysosomal binding partner of GATOR1 that acts either upstream or in parallel to the negative regulator [24,39]. The link between the regulation of mTORC1 activity and the perception of amino acid availability are cytosolic amino acid sensors. Sestrin2 and CASTOR1 directly interact with the amino acids, leucine and arginine, respectively, via a defined binding pocket [42,43] (Figure 4). During leucine starvation conditions Sestrin2 binds and inhibits GATOR2 function resulting in mTORC1 inhibition. In contrast, in the presence of leucine, the amino acid is bound by monomeric Sestrin2 triggering the dissociation of the sensor from GATOR2 enabling mTORC1 activity [42]. The cytosolic arginine sensor CASTOR1 acts in a similar fashion, by binding and inhibiting GATOR2 and therewith mTORC1 activity during arginine starvation conditions [43,44]. If arginine is present, two molecules of the amino acid are bound by homo-dimeric CASTOR1 causing its dissociation from GATOR2 relieving mTORC1 inhibition. The methionine status in mammalian cells however is sensed indirectly via the GATOR1-KICSTOR promoting protein SAMTOR [36]. Different from Sestrin2 and CASTOR, methionine’s metabolite SAM is sensed, instead of the amino acid itself (Figure 4). SAMTOR contains a highly conserved class I Rossmann fold methyltransferase domain, which functions as a binding pocket for SAM [36,45,46]. Methionine restricted growth conditions are reflected in the intracellular decrease of SAM levels [47]. Under these conditions, the SAM-binding domain in SAMTOR is unoccupied such that SAMTOR can interact with and promote the negative regulatory function of GATOR1-KICSTOR resulting in mTORC1 inhibition. In the presence of SAM, the metabolite disrupts the interaction of SAMTOR with GATOR1-KICSTOR promoting mTORC1 activity [36]. It is noteworthy that low abundance of folate and cobalamin can also be reflected in decreased SAM levels, hence SAMTOR might even be a link between the mTORC1 pathway and the availability of these vitamins [36]. 

SAMTOR orthologs are present in most animal kingdoms, interestingly none can be found in Saccharomyces cerevisiae [36]. Hence, the aforementioned PP2A methylation status is the only indirect methionine/SAM sensor in the mTOR pathway in yeast known today [30,34]. It is likely that other mechanisms that monitor methionine metabolism and SAM availability also exist in mammals, because SAMTOR or mTORC1 do not appear to play a role in coordinating methionine metabolism with cell proliferation [47,48]

## 5. Methionine and Its Role in Aging

Throughout different eucaryotic species, caloric restriction appears to positively influence healthy aging and longevity [49,50,51,52]. It seems that these benefits are not simply due to reduction of the daily energy supply, but rather a result of the decrease in protein intake [50,53]. Strikingly, lifespan extension in budding yeast [54], fruit flies [55], nematodes [56], and mice [57,58] was reported when exclusively methionine was limited. Metabolic alterations such as decreases in inflammation [58,59] and adiposity [60,61,62,63], or increased insulin-sensitivity [60,64,65] are some of the positive benefits of methionine restriction (MR). While mechanisms of lifespan extension are very complex and not well understood, it is tempting to propose MR-induced changes in SAM availability as a contributing factor. Indeed, experiments in C. elegance demonstrated that reducing SAM synthesis results in lifespan expansion [66]. Consistent with a role of SAM levels in aging, a strong correlation between DNA-methylation of sets of CpG regions with aging has been established in mammals [67]. Many of these regions are hypermethylated in aged individuals and it is conceivable that MR leads to reduced cellular methylation potential to counteract hypermethylation associated with aging. MR-induced reduction in SAM levels could also benefit lifespan via SAMTOR-dependent inhibition of mTORC1 [36]. There are several potential mechanisms that link MR to longevity, and given the complexity of the aging process it is expected that multiple pathways contribute. 

Intriguingly, MR also decreases oxidative stress, even though methionine is indirectly consumed to generate the antioxidant GSH [68,69,70,71]. In fact, overall plasma GSH levels increased due to alterations in sulfur-amino acid metabolism, when rats are fed a methionine restricted diet [71,72,73]. The extension of lifespan is likely promoted by the aforementioned metabolic changes and their combined benefits. In addition, MR-related longevity also possibly results from reduced incidences of cancer and overall reduction in cancer mortality [73].

## 6. Methionine and Its Role in Cancer

Cancer cells show a higher demand for nutritional supplies and certain metabolites compared to normal cells [4,74]. As mentioned above, the methionine metabolism is tightly regulated and coupled to other metabolic cycles (Figure 1). Therefore, it is not surprising that the activity or abundance of certain enzymes involved in these pathways can be altered in some types of cancer. Some tumor-initiating cells show a tendency for increased activity of enzymes involved in the methionine metabolism, MAT2A, the enzyme required to covert methionine to SAM under ATP consumption, is one of them [75]. Elevated activity of MAT2A suggests an increase in SAM production and methylation events resulting in high methionine-dependency in these cells. Indeed, transient pharmacological inhibition of the methionine cycle at the level of MAT2A seems to efficiently block tumor-initiation [75]. How some tumors increase MAT2A levels is not well understood, because MAT2A expression is tightly controlled by a feedback loop that links the cellular methylation potential with translation of MAT2A. Methylation of the MAT2A RNA reduces its translation thereby generating a self-sustaining control mechanism of MAT2A abundance and SAM synthesis [4,76,77].

Compared to normal cells, most cancer cells show an altered methylation pattern with a trend of overall decreased DNA methylation, however significant hypermethylation of specific genes [78]. Those increased methylation events can lead to the silencing of genes [79], which can be detrimental if the affected gene encodes a tumor-suppressor. Since methylation is a reversible reaction in the cell, and the predominant methyl donor SAM is generated from methionine, it is suggested that MR might suppress such hypermethylation hot spots in some types of cancer.

Alterations in the pathway of polyamine synthesis were also reported in some cancer types. Adenosylmethionine decarboxylase (AMD1) converts SAM to dcSAM and can be seen as a link between methionine cycle and polyamine synthesis [4,80]. The expression of AMD1 appears to be controlled by mTORC1 and was shown to be upregulated in certain prostate cancer types [80]. SAMTOR-mediated inhibition of mTORC1 might affect AMD1 expression levels and thus influence polyamine synthesis.

The requirement of methionine for cancer cell proliferation was already recognized over 60 years ago [81] and since then further defined by showing that most cancer cells are not able to proliferate when methionine in the growth media is substituted with homocysteine [48,82,83,84,85,86]. However, most cancer cell lines are able to readily metabolize homocysteine and convert it into methionine [83,87]. The dependency on supplied methionine seems to stem from an increased demand for methionine derived metabolites. Non-cancer cells, however, do not show changes in their ability to divide and grow in homocysteine media [47,48,85,87]. This methionine-dependency of cancer cell proliferation, which cannot be rescued by its metabolic precursor homocysteine is often referred to as the Hoffman effect. Methionine-dependency of cancer is likely a reflection of a general metabolic cell cycle checkpoint that responds to SAM availability. All cells from simple unicellular yeast to mammalian cells trigger this checkpoint arrest when methionine or SAM is limiting. Cancer cells seem to be hypersensitized due to their increased demand on fueling methionine metabolism and thus experience the Hoffman effect. 

## 7. Methionine Metabolism and Cell Cycle (SAM Checkpoint)

To ensure cellular and genomic integrity during stress conditions, cells are able to pause their cell cycle at specific checkpoints. Once favorable conditions are restored by specific stress response programs, the cell cycle can progress. If cells remain in the arrested stage for extended periods, apoptosis will usually be initiated. Reduction in the cellular methylation potential (SAM/SAH) is readily detected and leads to rapid induction of cell cycle arrest. This is likely a protective measure to preserve cellular, epigenetic, and genetic integrity, and is referred to as SAM checkpoint. 

As outlined in a previous section, in yeast the abundance of methionine or SAM is sensed by the ubiquitin ligase SCF^Met30^ and connects nutrient availability with cell proliferation through the Met4/Met32 regulators. Met32 stabilization blocks the initiation of S-phase resulting in a G1-phase arrest. Mechanistically it is not clear how Met32 inhibits cell cycle progression. However further downstream in this pathway, disassembly of pre-replication complexes (preRCs) appear to play a key role in the G1-phase arrest [88]. preRCs are essential for cell cycle progression as they provide a scaffold platform for the DNA replication machinery for transition into S-phase. Hence, preRC dissociation from chromatin upon methionine starvation is leading to a G1-phase arrest [88]. This process is evolutionary conserved, as destabilized preRCs also prevent S-phase initiation in most cancer cell lines grown in homocysteine media. Moreover, Cdc6, which is required for preRC assembly, was not only less abundant but also shown to dissociate from DNA when cells are grown in homocysteine media or when methionine was depleted from non-cancer cells [48] (Figure 5).

In mammals the SAM-checkpoint arrest at the G1-phase is reinforced by reduced activity of cyclin dependent kinase 2 (Cdk2) under methionine limiting conditions [48]. When grown in homocysteine media, Cdk2 is hypo-phosphorylated on the activating threonine-160 site, which results in reduced kinase activity [48,89]. The decrease in Cdc6 levels may contribute to threonine-160 hypo-phosphorylation as it is required for efficient modification of this site [90]. Stimulation of mitogen-activated kinase p38 and its downstream substrate MAP kinase MK2 are also part of the cascade that coordinates the SAM-checkpoint cell cycle arrest [47]. Overall, there are several parallels between the cell cycle events that are coordinated by methionine metabolism in yeast and mammals. Methionine or SAM limitation leads to cell cycle arrest at the G1/S transition, by blocking initiation of DNA replication, likely through loss of prereplication complexes from chromatin [48,88]. However, signaling cascades that initiate the arrest appear to be restricted to the SCF^Met30^ pathway in yeast, whereas MAP kinase signaling may be the dominant signal in mammals. What factors sense methionine or SAM abundance in mammalian cells in the context of the Hoffman effect or the SAM checkpoint is currently unknown. Even though methionine depletion induces a SAMTOR-dependent inhibition of TORC1 [36], mTOR signaling is unaffected when methionine is replaced with homocysteine, yet a robust cell cycle arrest is induced [48]. Furthermore, constitutive mTOR signaling is unable to override cell cycle block induced by methionine or SAM limitation [47], likely excluding SAMTOR as factor of the SAM checkpoint. Other sensors likely exist that link methionine metabolism with cell proliferation. Whether these are reminiscent of pathways in yeast remains to be investigated.

## 8. Concluding Remarks

Methionine metabolism is connected with several important pathways that influence cellular redox states, methylation potentials, phospholipid balance, as well as nucleotide and polyamine synthesis (Figure 1). Several sensing and signaling pathways have been discovered that report the status of methionine metabolites to other cellular processes, most notably cell proliferation and cell death. The first in line when it comes to the sensing of methionine or SAM in yeast is the E3 ligase SCF^Met30^ (Figure 2), which orchestrates the major response program to restore methionine levels and therewith methylation potential. So far, an element in the methionine perception pathway that actively binds the amino acid or its derivates similar to SAMTOR has not been identified. However, unpublished data in our lab indicate that a methionine or SAM sensing element may be located in the N-terminal region of F-box protein Met30. SCF^βTRCP^, the mammalian homolog of SCF^Met30^ has not yet been connected to coordination between proliferation and methionine abundance. It is also possible that other mammalian F-box proteins serve as functional homologs of yeast Met30 in this pathway. Further research is required to investigate the involvement of ubiquitin ligases in mammals, which is likely considering the central role SCF^Met30^ plays in the yeast system. The involvement of an E3 ligase in this pathway would present great potential for pharmacological approaches. The ubiquitin proteasome system has been successfully used for drug targeting in cancer therapy and other age-related diseases [91]. 

The TORC1 signaling pathway is influenced by the abundance of methionine or SAM levels in both, yeast and mammals (Figure 3 and Figure 4). Under normal growth conditions, yeast PP2A is activated by methylation of its catalytic subunit by Ppm1. Active PP2A suppresses SEACIT by dephosphorylation of Npr2p, which maintains TORC1 activity and results in inhibition of autophagy. If the methylation potential in yeast is reduced, PP2a is inactivated and Npr2p phosphorylation is promoted. Downstream this results in the inactivation of TORC1 and therewith induction of autophagy. In mammalian cells, PP2A was also shown to be methylated by the SAM-dependent leucine carboxyl methyltransferase 1 (LCMT1) [92,93]. Further, SAM-induced PP2a methylation promotes mTORC1 activity and suppresses autophagy in mammals [94]. However, it is not clear if PP2A facilitates dephosphorylation of NPRL2 (Npr2p in yeast), a component of GATOR1, the mammalian homolog of SEACIT. Further research will be required to test the role of PP2A methylation in sensing methionine levels in mammalian cells and the contribution of changes in PP2A methylation levels to control cell proliferation. 

While we have discovered some of the components that mediate communications between cell proliferation and methionine metabolism, much remains unknown, especially in the context of the Hoffman effect, which demonstrates that cancer cells rely on exogenous methionine. In preclinical models, a variety of different cancer types showed significantly decreased tumor growth when dietary methionine was restricted [95,96,97,98,99]. When MR is combined with chemotherapy or radiation, tumors become more sensitive towards the treatment [5,95] (Hoffman 2019, Gao 2019). Hence, dietary restriction of methionine in combination with classic treatment approaches seem very promising to clinical application. Continued identification and mechanistic characterization of the elements that sense methionine or SAM abundance and coordinate this metabolic pathway with cell proliferation will however be important to uncover therapeutic targets and biomarkers to identify susceptible cancer types and patients.

## Figures and Tables

**Figure 1 metabolites-11-00083-f001:**
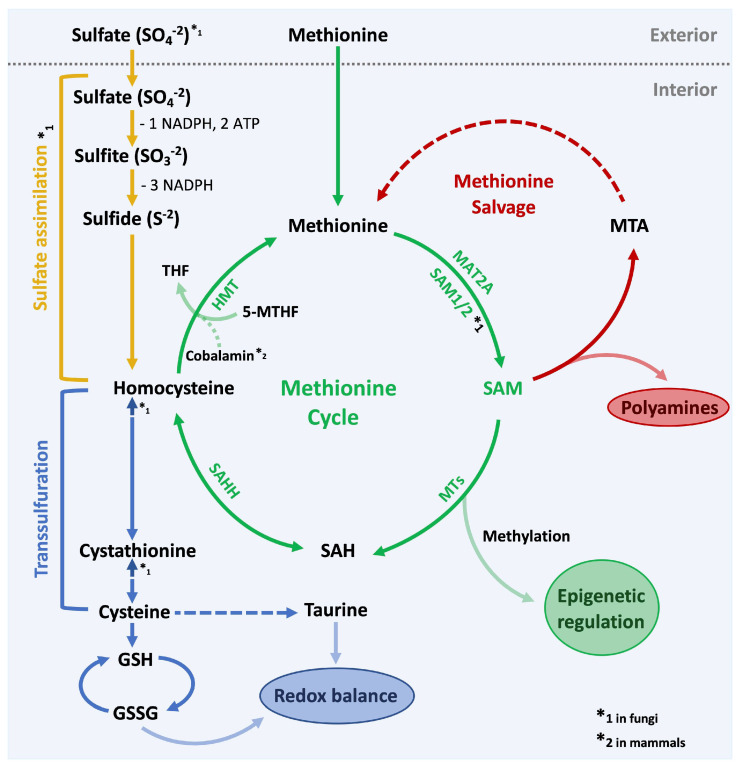
Methionine metabolism. Overview of the methionine cycle and other tightly coupled metabolic pathways. In green the methionine cycle, which quickly converts methionine to S-Adenosylmethionine (SAM) to generate methylation potential for the cell. Enzymes in the methionine cycle: MAT2A (methionine adenosyl transferase), SAM1/2 (SAM synthetase), Methyltransferases (MTs), SAHH (SAH hydrolase), HMT (Homocysteine methyltransferase). SAM is the primary methyl-group donor and is required for epigenetic regulation and other methylation-controlled processes. In red the methionine salvage pathway: SAM is also required for the synthesis of polyamines. Byproducts of this pathway are recycled to regenerate methionine. In blue the transsulfuration pathway: Homocysteine is converted to cysteine, which feeds into the generation of glutathione (GSH) and taurine to maintain the redox balance in the cell. In yellow the sulfate assimilation pathway: Most fungi are able to generate methionine from absorbed sulfate. This process is energy demanding and consumes 2 ATP and 4 NADPH molecules per one molecule of methionine.

**Figure 2 metabolites-11-00083-f002:**
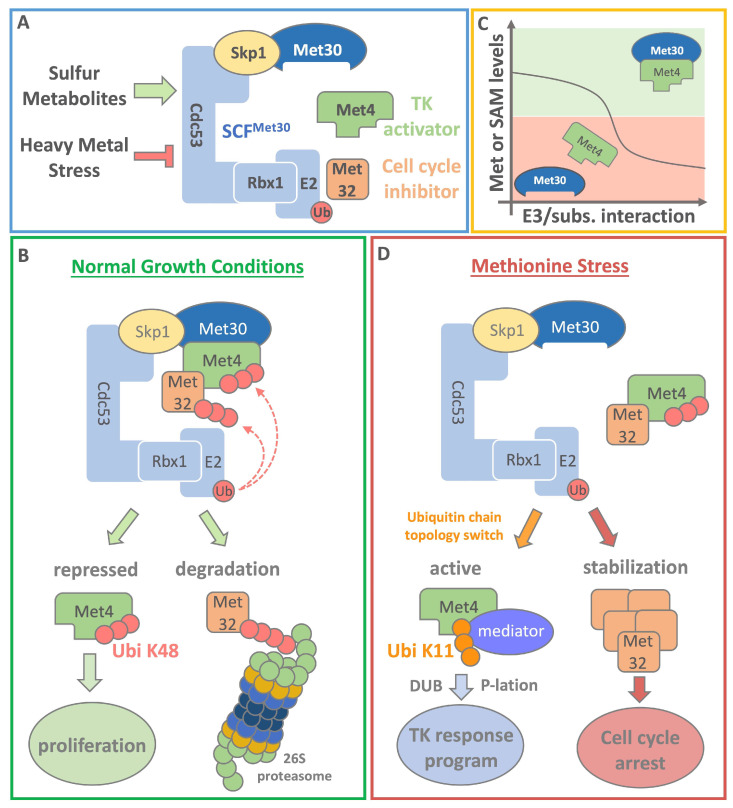
Methionine perception in yeast. (**A**) Blue box: The E3 ubiquitin ligase SCF^Met30^ is the center of the response to methionine starvation and heavy metal (cadmium) stress in yeast. The most critical substrates of the E3 ligase are the transcriptional activator Met4 and the cell cycle inhibitor Met32. Together they orchestrate the transcriptional response program during nutritional or heavy metal stress, however through profoundly distinct mechanisms. (**B**) Green box: Under normal growth conditions Met4 is ubiquitinylated by SCF^Met30^. The attached K48 linked ubiquitin chain does not lead to the degradation of Met4. Instead, Met4 is kept in a stable and transcriptionally inactive state. Met4 is also a substrate receptor for the cell cycle inhibitor Met32. Met4 recruits Met32 to SCF^Met30^ to catalyze the ubiquitylation and subsequent degradation of the cell cycle inhibitor via the 26S proteasome pathway to ensure proliferation. (**C**) Yellow box: Under normal growth conditions, high levels of methionine and metabolites promote a stable Met4 complex with the E3 ligase via the F-box protein Met30. Low methionine or SAM levels lead to the dissociation of SCF^Met30^ from its substrate Met4. (**D**) Red box: Under methionine or SAM starvation conditions, a change in Met4 ubiquitin chain topology is induced from K48 to K11 linkage, allowing the association of the transcriptional coactivator complex mediator with Met4. After this initiation step, Met4 is deubiquitylated (DUB) and phosphorylated (P-lation). The now active Met4 drives starts a transcriptional (TK) response program that drives genes to restore SAM levels. Dissociation of Met4 in turn also blocks ubiquitylation of Met32, leading to its stabilization. Accumulation of Met32 triggers a cell cycle arrest. Once sulfur containing metabolites are restored binding between SCF^Met30^ and its substrates is promoted. Met4 and Met32 get re-ubiquitylated resulting in transcriptional inactivation and degradation respectively. The cell cycle arrest is terminated and proliferation can progress.

**Figure 3 metabolites-11-00083-f003:**
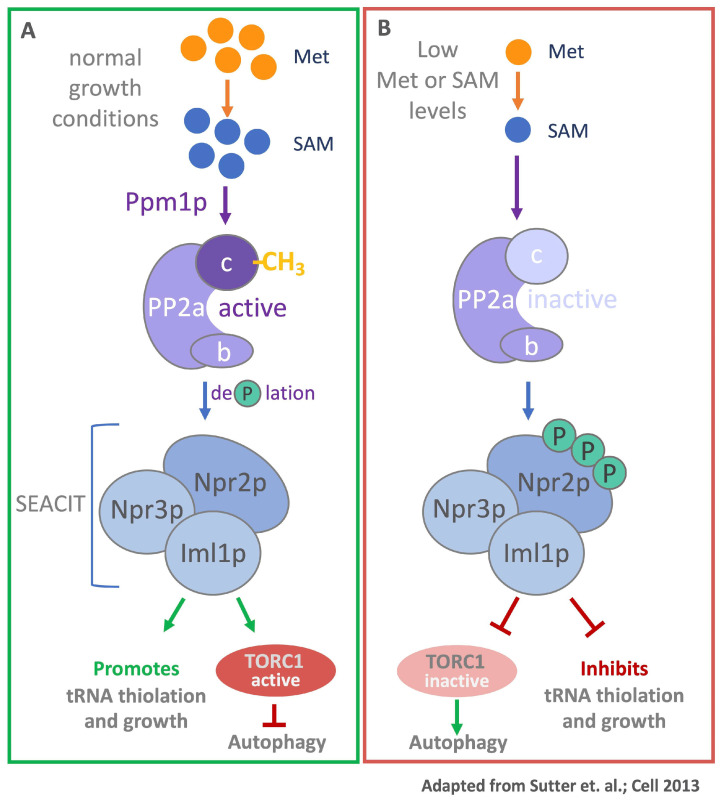
SAM-responsive methylation of PP2a. (**A**) Under normal growth condition, when methionine and SAM are plentiful, PP2A is activated by methylation of its catalytic subunit by Ppm1p. Active PP2a dephosphorylates Npr2p, a component of the autophagy repressing complex SEACIT (SEAC subcomplex Inhibiting TORC1 signaling). Growth is promoted and autophagy is inhibited by active TORC1. (**B**) When methionine or SAM levels are low, PP2A is not methylated and therewith inactive, leading to hyperphosphorylated Npr2p. Under these conditions growth is repressed, TORC1 is inactive, and autophagy is promoted.

**Figure 4 metabolites-11-00083-f004:**
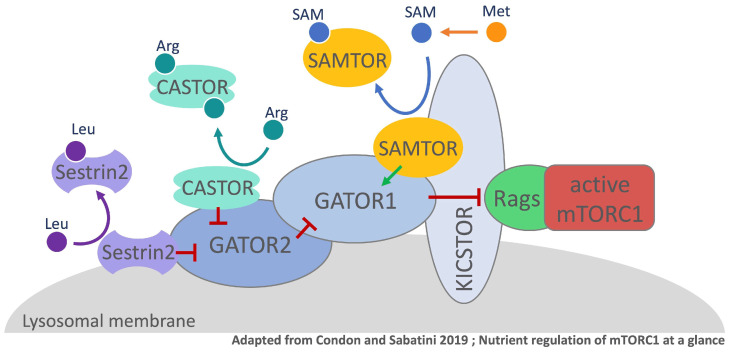
Amino acid perception in mammalian cells via mTORC1. For simplicity, only the components that are mentioned in this paragraph are shown in the model. mTORC1 (mechanistic Target of Rapamycin Complex 1), Rags (Rag GTPases), KICSTOR (KPTN-, ITFG2-, C12orf66-, and SZT2-containing regulator of TOR), GATOR1 (GAP activity towards the Rags 1), GATOR2 (GAP activity towards the Rags 2), CASTOR (Cellular Arginine sensor for mTORC1), SAMTOR (S-adenosylmethionine sensor upstream of mTORC1). mTORC1 activity coordinates availability of nutrients with cell growth. Certain amino acids are sensed in the cell and their abundance regulates mTORC1 activity. Leucine binds to its sensor Sestrin2, which leads to Sestrin2 dissociation from GATOR2. Leucine-starvation benefits the interaction of Sestrin2 and GATOR2 promoting the inhibition of mTORC1. CASTOR operates in a similar fashion, however binds 2 molecules of the amino acid arginine. During arginine starvation, CASTOR interacts with and inhibits GATOR2 resulting in subsequent mTORC1 inhibition. Methionine is rapidly converted into SAM and SAMTOR binds the metabolite. In the absence of SAM, SAMTOR interacts with KICSTOR and GATOR1 to inhibit mTORC1 signaling.

**Figure 5 metabolites-11-00083-f005:**
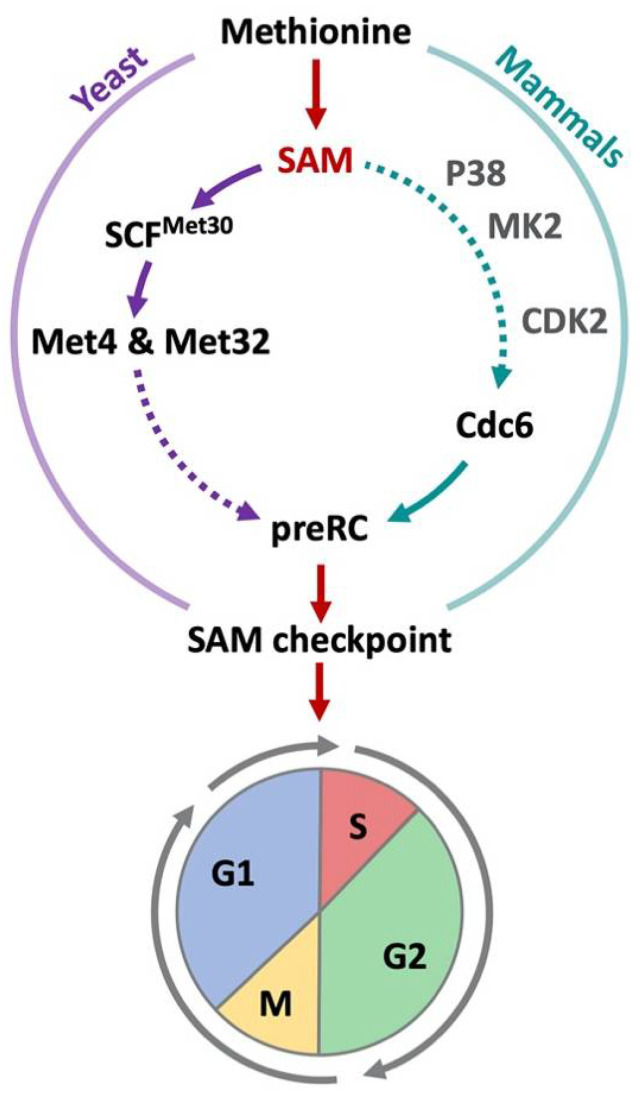
SAM-checkpoint in yeast and mammals. In yeast and mammalian cells, limiting methionine or SAM levels trigger a cell cycle arrest. In cancer cells, the arrest can also be induced when cells are shifted from methionine to homocysteine containing culture media (Hoffman effect). In both yeast and mammals, a destabilization of pre-replication complexes induces the SAM-checkpoint arrest. S-phase cannot be initiated and cells show an arrest in the G1-phase. Additionally, a delay in G2/M is observed. In yeast, sulfur containing metabolites are sensed by SCF^Met30^ and its substrates Met4 and Met32 coordinate methionine metabolism with proliferation. In mammalian cells, SAM is sensed by SAMTOR to regulate mTORC1 activity, yet its involvement in the SAM checkpoint is unlikely (see main text). However, p38, MK2, and Cdk2 have been suggested to play a role in the stability of pre-replication complexes. How this pathway receives information about methionine metabolism is however unknown.

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
