# Peer review of "Sensing and Signaling of Methionine Metabolism"

_metabolites, 2021, doi:10.3390/metabo11020083_

Round 1

Reviewer 1 Report

This is an interesting review on "methionine sensing" and "roles of methionine in aging and cancer" and is well-written in general. However, the link between these two is not explained enough. For example, it may be good to relate dysregulated proliferation and SAM checkpoint in cancer cells in terms of methionine metabolism. Aging/prolonged lifespan might be a different story.  

Other concerns:

  1. Mention (hypo/)taurine in the downstream of cysteine in the text and Figure 1. Modify Figure 1 legend as "Homocysteine is converted to cysteine, which feeds into the generation of glutathione (GSH) and (hypo/)taurine to maintain redox balance in the cells." Taurine is also related to longevity and health status in various species. Add 2ATP and 4 NADPH where needed during sulfate assimilation.
  2. Figure 2(/3): Use black letters (not white on pale backgrounds; hard to see Met4/Skp1 (Fig.2) and KICSTOR (Fig.3) on print) and add clear edges on all items for clear vision. "Met/SAM levels" is open to misunderstanding, Met plusSAM levels or Met/SAM ratios (as SAM/SAH ratio [line 188]). What is the mediator (a transcriptional coactivator complex in the manuscript but not legend), K11 orange circles (different from red circles=ubiquitin), DUB, TK, where are phosphorylated residues in Met4?
  3. It is hard to follow TORC1 signaling in yeast only by text (lines 168–218). What are different from mammalian mTORC1 signaling in Figure 3?

Author Response

Thank you for reviewing our manuscript. 

We agree it is correct and mechanisms are almost certainly different. We separated aging and cancer into two separate parts.

Other concerns:

1) The main text, Figure 1 and its legend were updated and the suggestions were integrated.

2) Thank you for pointing out, that the contrast of the initially used colors and lack of edges are hard to see on an actual print out. We changed some colors and added defined lines to the shapes and hope, this will improve figure 2. Additionally, we also edited figure 2 and its legend to address the other questions that were asked.

3) We added an additional Figure (Figure 3 = PP2A/TORC/autophagy; Figure 4 = SAMTOR) and hope the yeast pathway will be easier to follow. In addition, we now also discuss the known similarities and differences between yeast and mammalian methionine dependent mTORC1 activity in the concluding remarks.

Reviewer 2 Report

This manuscript on methionine metabolism by Lauinger & Kaiser provides a very comprehensive review in the field. The manuscript is well structured and written, with accurate description of the state-of-the-art. It raises the interest on the emergent field of methionine imbalance impact in cancer and other age-related diseases. I am supportive of its publication in the present form with minor revisions as suggested below.

  1. In the concluding remarks, the authors should sum up the unanswered questions raised during the manuscript (e.g. SCF^TRCP function in mammals and the direct sensor in yeast) besides the Hoffman effect. The authors should provide a personal view into the relevant questions to address in the future. 
  2. Figure 1 should include the enzymes involved in the Methionine Metabolism.
  3. Boxes in Figure 2 should be enumerated a) to d). Also, include description for the TK response program.
  4. Figure 3 should include the authophagy pathway description provided for yeast in lines 168-204.
  5. Check spacing in line 310 after 'types of cancer'.
  6. In line 346 replace cancer cells lines by cancer cell lines.

Author Response

Thank you for reviewing our manuscript.

We addressed all concerns and hope this will improve our manuscript.

1) We edited our concluding remarks and included a broader discussion concerning parallels in pathways and future prospects.

2) We added the key enzymes to the Methionine cycle in green, and agree this improves the figure. However, we decided not to add more enzymes to the other cycles to prevent the figure from being too crowded.

3) The figure was edited and the boxes are now labeled from A-D, we also updated the figure legend

4) Figure 3 is now a new addition which explains the mTORC regulation via PP2a methylation in yeast. The mammalian mTORC regulation via SAMTOR is now figure.

5) Was corrected

6) Was corrected 

Reviewer 3 Report

General: This review summarizes current knowledge of how methionine is perceived in yeast and mammalian cells and how methionine metabolism impacts cancer and aging. The authors cover details of methionine metabolic pathways, methionine sensing mechanisms and methionine metabolism’s role in aging/cancer and cell cycle arrest. Except for minor comments as noted below, the manuscript is clearly written, and each section is well described.

Comments:

  1. In Conclusion Remarks section, authors pointed out that future research would fill knowledge gaps and uncover novel therapeutic targets, without a relatively extended discussion on these future directions. Considering the purpose of writing a review is not only to summarize the current knowledge, more importantly, is to shed lights on future study, therefore the manuscript would be strengthened by a relatively extended discussion on the future research directions and potential therapeutic applications.
  2. Some definitions of the abbreviations are missed in the manuscript, e.g., SCF (Skp1, Cullins, F-box proteins), bZIP (Basic Leucine Zipper).

Author Response

Thank you for reviewing our manuscript.

We addressed all concerns and hope this will improve our manuscript.

1) We edited our concluding remarks and included a broader discussion concerning parallels in pathways and future prospects and agree.

2) The missing abbreviations were added.

Round 2

Reviewer 1 Report

The paper has been improved.

Reviewer 3 Report

The authors have addressed my concerns. I recommend acceptance of the revised version of the manuscripts.